# Characterization of Sub-Resonant Dielectric Spheres by Millimeter-Wave Scattering Measurements

**DOI:** 10.3390/s25185687

**Published:** 2025-09-12

**Authors:** Max Lippoldt, Jan Hesselbarth

**Affiliations:** Institute for Radio Frequency Technology (IHF), University of Stuttgart, 70569 Stuttgart, Germany

**Keywords:** electromagnetic scattering, inverse scattering, Mie theory, millimeter-wave

## Abstract

**Highlights:**

**What are the main findings?**
Sub-wavelength-sized alumina ceramic spheres are characterized with good accuracy with respect to their size and relative permittivity from scattering measurements at millimeter-wave frequencies.The characterization procedure is explained in detail and systematic error sources are analyzed.

**What are the implications of the main findings?**
Sub-resonant scattering measurements allow the characterization of relatively smaller particles with respect to traditional resonance-based methods.Alternatively, for a given particle size, sub-resonant characterization allows use of a lower operating frequency and, thus, simpler equipment.

**Abstract:**

When measuring the size or relative permittivity of a dielectric particle, usually one of the parameters needs to be known for determining the other one. Scattering measurement methods offer an alternative that allows for extracting both the size and the permittivity of the particle under test at the same time. In this paper, bi-static scattering measurements at millimeter-wave frequencies are applied to characterize sub-resonant dielectric spheres of sub-wavelength size. The size and relative permittivity are extracted simultaneously from measurements at Ka-band (26.5–40 GHz). The experimental setup and the data processing procedure are detailed in depth, and the sources of the systematic errors are discussed. Alumina ceramic spheres (with relative permittivity of approximately 10) of diameter as small as 0.8 mm (less than 1/10 of free-space wavelength) were investigated. The extracted diameters and permittivities agree well with the expected values.

## 1. Introduction

The scattering of electromagnetic waves by a spherical particle has been of interest to the scientific community for a long time [1,2]. Commonly, its analysis is attributed to Mie [3] and is referred to as the Mie theory. Initial research explored the so-called direct scattering problem: For a given particle, calculate its scattering to explain the observed phenomena. In this way, Mie explained the scattered color spectrum of colloidal gold solutions [3].

The focus of this paper is on the inverse problem: from the measured scattering, find the parameters of the scatterer. Generally, this requires some previous knowledge about the scatterer’s shape and material class (e.g., lossy metal, nonmagnetic dielectric material, etc.), to make the characterization unique [4] (p. 168). Inverse scattering has been widely used for sizing of particles of known material [5,6,7], even including shape determination [8,9], or for determining the permittivity of spheres of known size [10,11,12,13].

The Mie theory was originally developed for the optical spectrum, but microwave scattering experiments also have a long history [14]. Microwave measurements offer the advantage that the phase information of the scattered signal is retrieved easily, compared to optical measurements. Microwave experiments have been used to verify theoretical scattering calculations and simulations [15,16,17,18,19], including special scattering states (Kerker conditions) [20], for scattering measurements of complex aggregates that are equivalent to interstellar dust [21,22], or to study the influence of a particle’s shape on its scattering [23,24].

The focus of this paper is on the simultaneous extraction of the size and permittivity of dielectric spheres. This has been realized before for resonant dielectric spheres [25,26,27]. “Resonant” in this context means that the measured scattering intensity over frequency contains peaks that correspond to the resonant frequencies of the sphere’s eigenmodes [28]. In the following, however, the characterization of relatively smaller sub-resonant particles is explored and, for the first time (to the best of the authors’ knowledge), demonstrated at millimeter-wave (mm-wave) frequencies. “Sub-resonant” means that there are no resonant scattering intensity peaks in the measured frequency spectrum and that the measured scattering is relatively weak, thus making it much harder to characterize the sphere under test. While resonant methods, such as in [25], depend on the resonant frequencies of the measured intensity peaks to characterize the particle, this method is not applicable in the sub-resonant range. In contrast, the method used in this paper relies on a best-fit approach. It is generally similar to the one used in [27] for resonant spheres, but it is applied here at sub-resonant sphere sizes.

Working in the sub-resonant range allows us to characterize smaller particles at a given frequency. For example, scattering experiments at terahertz frequencies [29,30] if extended to sub-resonant scatterers would include scatterers of a few tens-of-micrometers size, reaching the size of interesting biological cells [31]. Alternatively, for larger particles, sub-resonant characterization allows us to use a lower operating frequency and, thus, simpler equipment.

This paper proposes a wide-band measurement and extraction scheme, which eases the limitations imposed by the finite dynamic range of the instrumentation. Additionally, systematic error sources are discussed, such as the effect of a height offset of the sphere under test and the necessity to include a scheme based on higher-order modes in the extraction procedure.

The paper is structured as follows. Section 2.1 reviews the Mie scattering of a dielectric sphere. Section 2.2 introduces the measurement setup. Section 2.3 explains the post-processing of the measured data. Section 3 reports the results extracted from the measurements and gives a conclusion as well as an outlook on future work.

## 2. Materials and Methods

### 2.1. Scattering of a Dielectric Sphere

The electromagnetic scattering of a homogeneous dielectric sphere suspended in free space, illuminated by a plane wave, can be described as [32] (p. 112)(1)$$\left(\begin{array}{c} E_{||s}\\ E_{⊥s} \end{array}\right)=\frac{e^{jk(r-z)}}{-jkr}\left(\begin{array}{cc} S_{\left|\right|} & 0\\ 0 & S_{⊥} \end{array}\right)\left(\begin{array}{c} E_{||i}\\ E_{⊥i} \end{array}\right)$$

Here, the scattered (*s*) and incident (*i*) electric fields are divided into a parallel and a perpendicular part with respect to the so-called scattering plane. In the following, referring to Figure 1, the scatterer is in the origin of the coordinate system and the scattering plane is the xz plane. The scattered fields are found from the incident fields by multiplication with the so-called amplitude scattering matrix (S). For a homogeneous isotropic sphere, there is no cross-polarization component of the scattering. The complex matrix elements $S_{⊥}$ and $S_{\left|\right|}$ can each be found as an infinite series of the Mie scattering coefficients $a_{i}$, $b_{i}$ [32] (p. 100, 112). These coefficients correspond to the electric or magnetic resonance modes of the sphere, meaning that $a_{1}$ belongs to the first electric dipole mode and $b_{1}$ to the first magnetic dipole mode. For the numerical calculation, the infinite series are truncated; the resulting errors are discussed in Section 2.3.

Notably, the Mie theory suffers from somewhat misleading terminology when applied to microwaves. The amplitude scattering matrix (S) with its elements (the amplitude scattering parameters $S_{⊥}$, $S_{\left|\right|}$), the Mie scattering coefficients $a_{i}$, $b_{i}$ (representing the resonance modes), and the transmission scattering parameters $S_{12}$, $S_{21}$ measured by the vector network analyzer (VNA) all have very similar names. In the following, this paper uses the term “scattering parameters of the sphere” for $S_{⊥}$, $S_{\left|\right|}$, the term “Mie coefficients” for $a_{i}$, $b_{i}$, and the term “transmission” for $S_{12}$, $S_{21}$ as measured by the VNA. All these are complex numbers and frequency-dependent.

A non-magnetic isotropic dielectric sphere is described by two parameters: size (radius *a*) and material (complex $\varepsilon_{r}$). Since the sub-resonant scattering shows very little dependence on a small dielectric loss of the scatterer (this being the case here for alumina ceramic), the imaginary part of the permittivity is neglected in the following. In the Mie theory, the information about the sphere is contained in the size parameter *x*, with x=ka, where *k* is the free-space wave number, and the contrast parameter $m=\sqrt{\varepsilon_{r}}$ (for a sphere in free space).

This paper considers alumina dielectric spheres ($\varepsilon_{r}\approx$ 10) in the sub-resonant regime. For dielectric spheres, the limit to the sub-resonant regime is the first magnetic dipole resonance, which can be approximately calculated by [33](2)$$f_{MD}\;\left[GHz\right]=\frac{149.9}{a\;\left[\mathrm{mm}\right]\;\times \sqrt{\varepsilon_{r}+2}}.$$

For the largest alumina sphere with radius 1 mm (see Figure 2), the magnetic dipole resonance accordingly occurs at 43.3 GHz. Consequently, as all the measurements are done in the Ka band (26.5–40 GHz) all the spheres used in this paper are sub-resonant.

### 2.2. Measurement of Bi-Static Sphere Scattering

As shown in Figure 3, the measurements are performed with a Keysight N5225B VNA connected to two rectangular waveguides (WR-28) with flared ends. The waveguides are turned manually to measure parallel and perpendicular polarizations. With the flare ends at a distance of 80 mm to the sphere (about $7\lambda_{0}$ at 26.5 GHz), far-field conditions apply. The sphere under test was placed on a holder made of Rohacell foam.

Figure 4 shows the signal path model of the transmissions measured by the VNA. The desired signal path is characterized by the sphere’s to-be-measured $S_{⊥}$, $S_{\left|\right|}$ together with unknown path transmissions and feed transmissions. There is also an unavoidable direct coupling path. Three measurements are necessary to determine $S_{⊥}$, $S_{\left|\right|}$ (cp. [34,35]). First, a known calibration sphere (here, a 1 mm radius stainless steel sphere) is measured. Second, the transmission without any sphere is measured. Third, the transmission with the sphere under test is recorded.

The scattering parameter of the sphere under test is then found by (cp. Figure 4, $S_{⊥}$ or $S_{\left|\right|}$, according to the measured polarization, for each frequency point):(3)$$S_{⊥,||}^{meas}\;=C\;\left(S_{21,test\;sphere}-S_{21,no\;sphere}\right)$$

The frequency-dependent calibration factor *C* is found at each frequency point as the ratio of the measured $S_{⊥}$ or $S_{\left|\right|}$ (with C=1) for the known calibration sphere divided by the theoretically calculated value (obtained from the Mie theory) for that sphere.

An accurate alignment of the measurement setup is essential and is achieved as follows: the height (*y*) position of the Rohacell holder is adjusted using a line laser so that the calibration sphere is in plane with the centers of the flared waveguide ends. A cross-line laser mounted over the measurement setup is used to accurately establish the measurement angle. An accuracy of better than ±0.5° is estimated for the experimental setup. In the xz plane, the spheres are centered using a movable table and a digital microscope.

Each VNA measurement of a sphere scattering is immediately followed by a related measurement of the transmission without any sphere. This procedure has the advantage of minimizing the effect of VNA drift. Removing small error effects wherever possible is important because the difference in measured transmission between the cases with a test sphere and without a sphere is very small, e.g., only $\left(5.1+j6.5\right)\cdot 10^{-5}$ for the 0.6 mm-radius alumina sphere at 30 GHz, which corresponds to a magnitude difference of roughly 0.11 dB and a phase difference of 0.17° at a transmission magnitude of approximately −44 dB.

A scattering angle of θ= 120° is chosen in this paper. Note that for θ= 0° and θ= 180°, no additional information is obtained from a dual-polarized measurement, since, there, $S_{⊥}=S_{\left|\right|}$. At θ≈ 90°, $S_{\left|\right|}$ is minimum and, thus, most susceptible to noise. Backward directions (90°<θ<180°) are less affected by direct (not-scattered) transmission than forward directions. As a good compromise, θ= 120° is selected.

In general, the center point of the sphere under test is slightly different from the one of the calibration sphere (as they are placed on top of the same Rohacell holder, but have different radii), creating an offset in the *y* direction. The extraction procedure (see (Equation 3)), however, assumes that their centers are at the same location. A simplified theoretical analysis of a *y*-offset sphere assumes that the additional path length Δl generated by an offset Δy (compare Figure 5) translates into a phase offset Δφ of the measured scattering parameters of the sphere, while their amplitude is assumed to remain constant. For a distance of l/2 between the antenna aperture and the reference sphere center, the following equation for the phase shift of the *y*-offset sphere is obtained from geometrical considerations:(4)$$\Delta \varphi \;=\;\frac{2\pi }{\lambda_{0}}\left(\sqrt{l^{2}+4\;\Delta y^{2}}-l\right)$$

For l/2=80 mm and an offset Δy≤0.6 mm (corresponding to the radius difference between the largest and smallest sphere used in this paper), a phase offset of the measured scattering parameter of the sphere of Δφ≤0.32° is predicted at a frequency of 30 GHz. Measurements with a 1 mm-radius alumina sphere at different *y* positions (varying *y* by up to ±1 mm) do not show any measurable phase variation of $S_{⊥}$, $S_{\left|\right|}$ beyond the measurement noise. Consequently, the *y* offset generated by a varying radius of the sphere under test is neglected here.

### 2.3. Post-Processing and Parameter Extraction

As described in Section 2.2, the measured scattering parameters $S_{⊥}$, $S_{\left|\right|}$ of a sphere under test are found from the measured transmissions for each frequency. Thus, the measurements give four real-valued functions (of frequency), namely, Re{} and Im{} of both $S_{⊥}$ and $S_{\left|\right|}$. From those, radius *a* and permittivity $\varepsilon_{r}$ are obtained as follows.

For many combinations of values for *a*, $\varepsilon_{r}$ and for the measured frequency range (26.5–40 GHz) the “theoretical” scattering parameters $S_{⊥}$, $S_{\left|\right|}$ of a sphere are calculated from (truncated) series of Mie scattering coefficients. This assumes that the permittivity is constant over the considered frequency range (in accordance with [36]). For each specific pair of *a*, $\varepsilon_{r}$, four error functions (EFs, i.e., error square sums) are calculated by comparing each of the four functions obtained for the measured scattering parameters with the corresponding “theoretical” scattering parameter function. Exemplarily, the definition of the error function considering $Re\{S_{⊥}\}$ is given here; the three other error functions (for $Im\{S_{⊥}\}$, $Re\{S_{\left|\right|}\}$, $Im\{S_{\left|\right|}\}$) are defined in the same way:(5)$$EF_{Re\{S_{⊥}\}}\left(a,\varepsilon_{r}\right)\;=\;\sum_{f}{\left(Re\left\{S_{⊥}^{meas}\left(f\right)\right\}-Re\left\{S_{⊥}^{theory}\left(f,a,\varepsilon_{r}\right)\right\}\right)}^{2}$$

The sum of these four error sums gives a total error function (TEF) value for each specific pair of *a*, $\varepsilon_{r}$:(6)$$TEF\left(a,\varepsilon_{r}\right)\;=\;\log_{10}\left(EF_{Re\{S_{⊥}\}}\left(a,\varepsilon_{r}\right)\right)+\log_{10}\left(EF_{Im\{S_{⊥}\}}\left(a,\varepsilon_{r}\right)\right)+\log_{10}\left(EF_{Re\{S_{\left|\right|}\}}\left(a,\varepsilon_{r}\right)\right)+\log_{10}\left(EF_{Im\{S_{\left|\right|}\}}\left(a,\varepsilon_{r}\right)\right)$$

This approach preserves the small degree of orthogonality that is obtained from measuring both polarizations, i.e., both $S_{⊥}$ and $S_{\left|\right|}$. Figure 6 shows an example of the resulting total error function over a range of the parameters. The minimum of this total error function yields the best possible fit between the theoretically calculated and measured scattering parameters for all four curves (Re{} and Im{} of $S_{⊥}$ and $S_{\left|\right|}$) at the same time and corresponds to the best extracted values for *a* and $\varepsilon_{r}$.

Figure 7 shows an example of one of the measured scattering parameters (Im{$S_{⊥}$}) with the best-fit curve of the “theoretical” scattering parameter. Note that the ripples of the measured (blue) curve originate from the measurement noise of the VNA. Also shown, for illustration, are two “theoretical” curves based on different settings for *a* and $\varepsilon_{r}$.

Observation of the behavior of the total error function (Figure 6) indicates a rather shallow error minimum over the variation of the permittivity. Figure 8 displays the minimum trace of the total error function magnitude over $\varepsilon_{r}$. It shows that using a wider frequency range in the experiment and analysis resulted in a more pronounced minimum of the total error function. This means that a wider frequency bandwidth leads to more stable and reliable extraction results. In fact, a single-frequency extraction would be theoretically possible but would be very unstable in the face of measurement noise because of the low orthogonality of $S_{⊥}$, $S_{\left|\right|}$ with respect to *a*, $\varepsilon_{r}$. Inclusion of the frequency variation of the scattering adds another degree of orthogonality, thereby improving the robustness of the extraction.

Additionally, it can be observed from Figure 6 that an uncertainty in the extracted radius causes a much stronger variation of the total error function than an uncertainty (of similar relative amount) in the permittivity. This is supported by (Equation 2) for the first resonance frequency, which is inversely proportional to the radius and is inversely proportional to the square root of the sum of two plus the permittivity. The latter means that variations of the radius have a 4-times-stronger response than variation of a low permittivity ($\varepsilon_{r}=$ 2), and, still, a response about 2.4 times stronger than a permittivity of $\varepsilon_{r}=$ 10. In consequence, the extracted radius will be less sensitive to noise or statistic measurement errors than the extracted permittivity.

For sub-resonant spheres, it might seem intuitive to simply consider $a_{1}$ and $b_{1}$ in the scattering calculation, as these dipole terms strongly dominate the scattering in the sub-resonant region. However, this approximation has only limited accuracy for the sub-resonant range used here.

Notably, the orthogonality of $S_{⊥}$, $S_{\left|\right|}$ with respect to *a*, $\varepsilon_{r}$ disappears for deeply sub-resonant spheres ($\sqrt{\varepsilon_{r}}\;\;x\ll 1$). For such spheres, the scattering parameters of the sphere approximately only depend on the electric dipole coefficient $a_{1}$ [32] (p. 131):(7)$$S_{⊥}=\frac{3}{2}a_{1}\;,\;\;\;S_{\left|\right|}=\frac{3}{2}a_{1}\cos\theta$$

For $\sqrt{\varepsilon_{r}}\;\;x\ll 1$, the electric dipole coefficient $a_{1}$ can be approximated as [32] (p. 131)(8)$$a_{1}=-\frac{j2k^{3}a^{3}}{3}\frac{\varepsilon_{r}-1}{\varepsilon_{r}+2}.$$

Under this approximation, $S_{⊥}$, $S_{\left|\right|}$ are linearly dependent for each angle θ. Also, the radius and permittivity of the sphere under test no longer influence the scattering separately, but multiple combinations of *a*, $\varepsilon_{r}$ exist that generate exactly the same scattering. This explains the shallow “minimum valley” found in the total error function in Figure 6. From these considerations, it follows that only the influence of higher-order modes (such as the magnetic dipole mode, the electric and magnetic quadrupole modes, etc.) makes the characterization of *a*, $\varepsilon_{r}$ unique and leads to a distinct minimum of the total error function. Consequently, a sufficient number of higher-order modes needs to be considered in the extraction procedure. Also, this means that the characterization of deeply sub-resonant particles would be very challenging, as that would require very precise measurements of the scattering to reveal the small contributions by higher-order modes in addition to the electric dipole mode.

In principle, the Mie theory calculates the scattering of a sphere with an infinite series that includes the scattering contributions of all spherical resonance modes. For numerical calculations, this series needs to be truncated. This gives an accurate approximation of the scattering if an appropriate number of higher-order modes are considered for a given sphere size. A criterion for truncation used in the literature is that of Wiscombe [37]:(9)$$n_{stop}\;=\;round\left(x+4x^{1/3}+2\right),$$
where $n_{stop}$ is the number of terms after which the infinite series is truncated. According to that criterion, for a sphere with size parameter 0.8 (sub-resonant for alumina), the first seven terms and, thus, the first 14 modes of the sphere should be considered, i.e., the Mie coefficients up to and including $a_{7}$, $b_{7}$.

Neglecting the Wiscombe criterion and using only the dipole coefficients $a_{1}$, $b_{1}$ to calculate the scattering parameters of a sphere would, in principle, allow the extraction procedure to work but can result in a significant error, as shown by the example in Figure 9. Only for very small spheres (x<0.2) would using only $a_{1}$, $b_{1}$ be a valid approximation. However, in the present case—sub-resonant but close to the resonance (0.2<x<0.8)—it is necessary to consider higher-order modes according to the Wiscombe criterion.

## 3. Results and Discussion

### 3.1. Results

The extracted results for all the spheres measured in this paper are shown in Table 1. As can be seen, the extracted radii agree well with the mechanically measured values. The extracted permittivities agree reasonably well with the literature values [36,38], although a trend toward increased permittivity with smaller sphere sizes can be observed. This might be due to a non-identified systematic measurement error and needs to be explored in the future.

### 3.2. Discussion

By using bi-static scattering, the simultaneous characterization of size and permittivity was successfully demonstrated for sub-resonant dielectric spheres at mm-wave frequency. The extracted results from the scattering measurements agree well with the expected values. Alumina ceramic dielectric spheres with a radius as small as 0.4 mm were characterized using Ka-band frequencies.

Systematic error sources were discussed, and we tried to eliminate them. Nevertheless, the obtained values for the permittivity hint at additional, non-identified error contributions. Moreover, the experimental system is delicate. Possible ways for future improvement include using antennas with higher gain and bringing the antennas closer to the scatterer, which will improve the dynamic range and possibly allow for the characterization of smaller scatterers with higher accuracy due to the better signal-to-noise ratio. However, this will also require the consideration of stronger direct couplings and effects, due to non-planar wavefronts. For very small distances between the antennas and the sphere under test, far-field conditions cannot be assumed and, thus, the Mie theory would no longer be applicable.

## Figures and Tables

**Figure 1 sensors-25-05687-f001:**
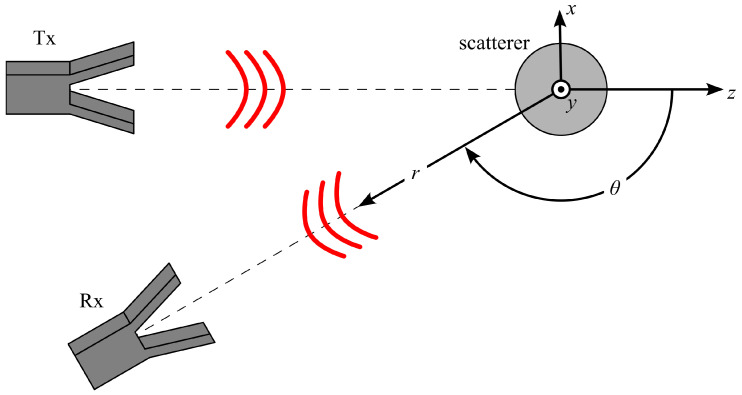
Geometry of the scattering setup, including the definition of the scattering plane (xz plane), the direction of the incident wave (+z direction), and the scattering angle θ.

**Figure 2 sensors-25-05687-f002:**
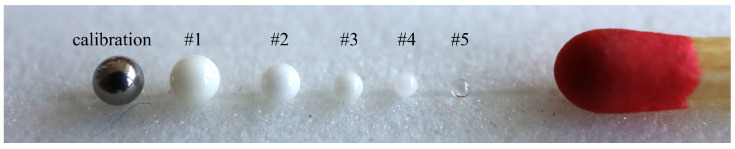
Spheres used in this work: metal calibration sphere (radius 1 mm) and alumina ceramic spheres (radii 1 mm, 0.794 mm, 0.6 mm, 0.5 mm, 0.4 mm). Match head for size comparison.

**Figure 3 sensors-25-05687-f003:**
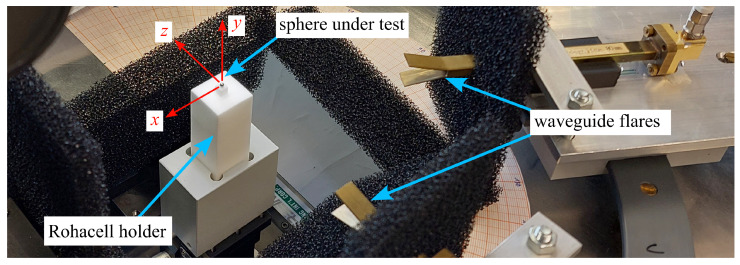
Photo of the measurement setup. The sphere under test rests on a Rohacell holder. Flared waveguides are used as antennas.

**Figure 4 sensors-25-05687-f004:**
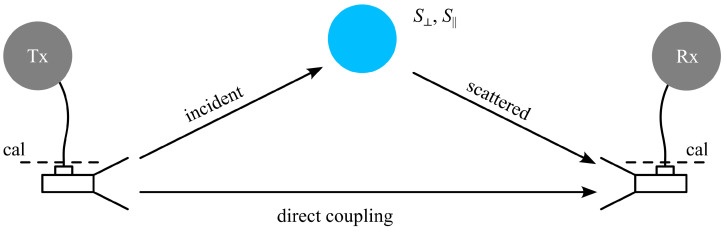
Signal flow model for the measured transmission. An unwanted direct coupling path is considered in addition to the wanted signal path that includes the scattering of the sphere together with an unknown path loss.

**Figure 5 sensors-25-05687-f005:**
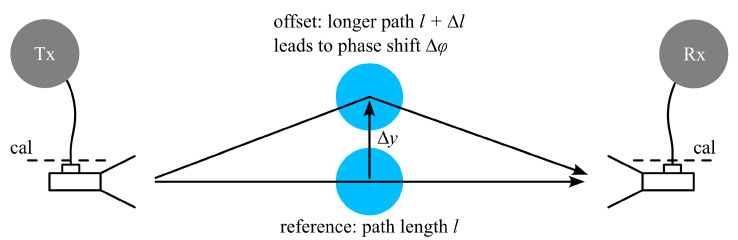
An additional path length due to a *y* offset of the sphere leads to a phase shift of the measured scattering parameter.

**Figure 6 sensors-25-05687-f006:**
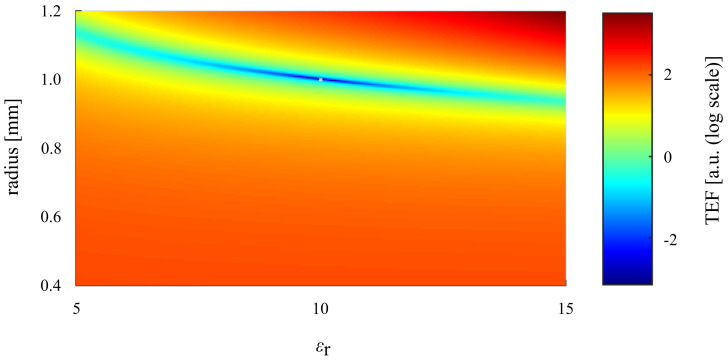
Illustration of the total error function over radius and permittivity (theoretically generated plot for a relative frequency bandwidth of 10%). At the minimum of the total error function, the best fit of the theoretically calculated and measured scattering curves (Re{} and Im{} of $S_{⊥}$ and $S_{\left|\right|}$ over frequency) is achieved. This gives the extracted radius and permittivity of the sphere under test.

**Figure 7 sensors-25-05687-f007:**
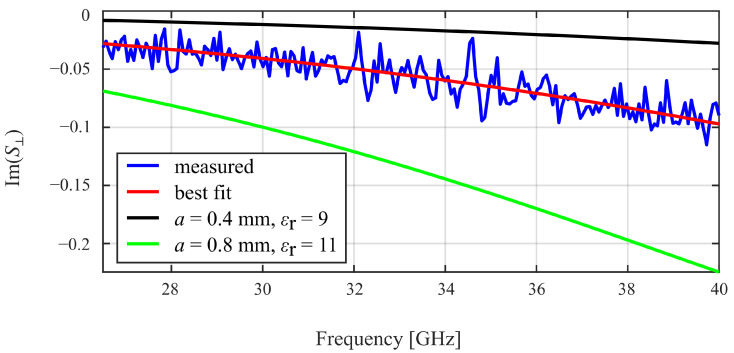
Measured scattering parameter Im{$S_{⊥}$} of a sphere under test, best-fitting “theoretical” curve, and two chosen curves of “theoretical” scattering parameters for different values of *a*, $\varepsilon_{r}$. By achieving the best possible fit, the sphere’s radius and permittivity are extracted (here: *a* = 0.596 mm, $\varepsilon_{r}$ = 10.633 for alumina ceramic sphere #3).

**Figure 8 sensors-25-05687-f008:**
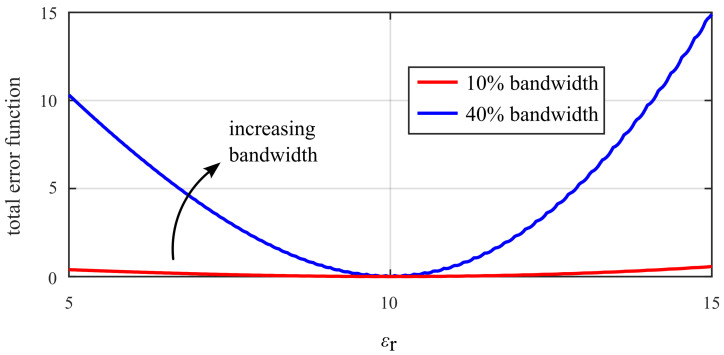
Minimum trace of total error function (tracing the bottom of the valley visible in Figure 6, varying radius) over $\varepsilon_{r}$ with increasing frequency bandwidth. A wider frequency bandwidth increases the steepness of the curve and, thus, makes the extraction less sensitive to measurement noise.

**Figure 9 sensors-25-05687-f009:**
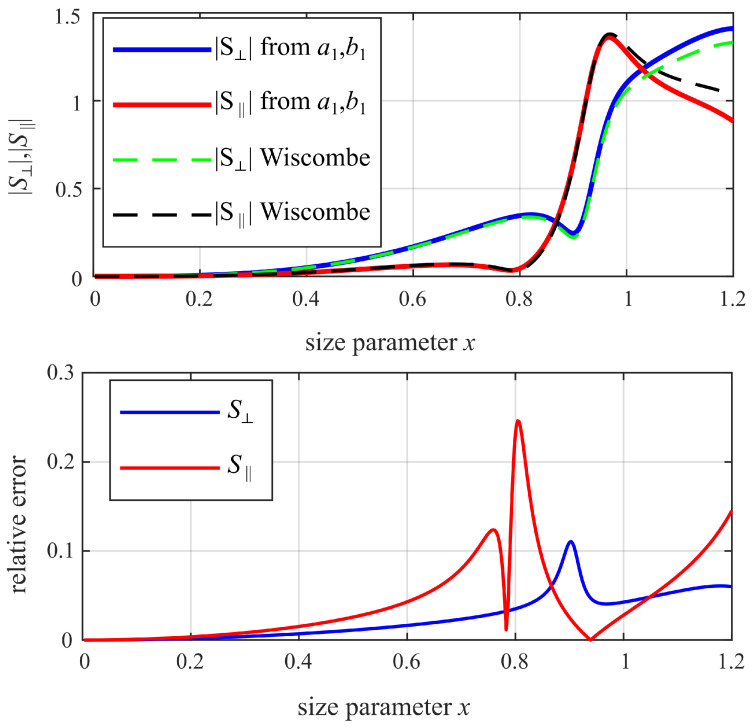
Magnitude of scattering parameters $|S_{⊥}|$, $|S_{\left|\right|}|$ versus size parameter (top) and relative error (bottom) when calculated using only the dipole coefficients $a_{1}$, $b_{1}$ in comparison to when calculated with higher-order modes according to the Wiscombe criterion (dielectric sphere with $\varepsilon_{r}$ = 10, θ=120°).

**Table 1 sensors-25-05687-t001:** Extracted results.

Sphere	Mechanically Measured Radius [mm]	Extracted Radius [mm]	Extracted $\varepsilon_{r}$
#1	0.998	1.006	9.967
#2	0.793	0.794	10.900
#3	0.599	0.596	10.633
#4	0.498	0.497	12.033
#5	0.396	0.394	12.867

## Data Availability

The original contributions presented in this study are included in the article. Further inquiries can be directed to the corresponding author.

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
