# Peer review of "Characterization of Sub-Resonant Dielectric Spheres by Millimeter-Wave Scattering Measurements"

_sensors, 2025, doi:10.3390/s25185687_

Round 1

Reviewer 1 Report

Comments and Suggestions for Authors

I understand the paper and feel that the authors did adequate job in presenting their work. I am recommending the work for publication although I feel that Resonance scattering is a better way to extract the required parameters.  I do not agree with advantages narrated by the authors. However, it is a alternate way of extraction and may be useful for other researchers.

Author Response

Reviewer 1
“I understand the paper and feel that the authors did adequate job in presenting their work. I am recommending the work for publication although I feel that Resonance scattering is a better way to extract the required parameters. I do not agree with advantages narrated by the authors. However, it is a alternate way of extraction and may be useful for other researchers.”

Answer:
As stated in the manuscript, working in the sub-resonant range allows to characterize smaller particles at a given frequency. Alternatively, for larger particles, sub-resonant characterization allows to use a lower operating frequency. Given the increased complexity of microwave equipment towards higher frequency, the authors do see a clear benefit from sub-resonant scattering techniques, working at comparably lower frequency, using lower-complexity (lower-cost) equipment. This comes at the trade-off with much weaker scattering response, compared to resonant scattering, as discussed in our research.

Reviewer 2 Report

Comments and Suggestions for Authors

please see the attached file. 

Author Response

Reviewer 2
“1. The distinction between "scattering parameters of the sphere", "Mie coefficients", and "transmission" is helpful but could be reinforced early in the introduction to avoid confusion.”

Answer:
We do not consider beneficial to mention this distinction in the introduction, as there the corresponding terms are not discussed yet. Too much technical discussion in the introduction may overload it and distracts from the topic.

“2. The outlook mentions using higher-gain antennas and reduced antenna-scatterer distance. Could the authors elaborate on expected improvements (e.g., SNR, resolution) and potential challenges (e.g., near-field effects)?”

Answer:
The outlook has been expanded accordingly (page 10, lines 260-267): “Possible ways for future improvement include using antennas with higher gain and bringing the antennas closer to the scatterer, which will improve the dynamic range and possibly allow for the characterization of smaller scatterers with higher accuracy due to the better signal-to-noise ratio. However, this will also require the consideration of stronger direct couplings and effects due to non-planar wavefronts. For very small distances between the antennas and the sphere under test, far-field conditions cannot be assumed and thus the Mie theory would no longer be applicable.”

“3. Please clarify the difference between the proposed method and the recently developed resonant imaging method, such as those presented in: A novel quantitative inverse scattering scheme using interior resonant modes, Inverse Problems 39 (8), 085002, 2023”

Answer:
Note that the mentioned paper is on the scattering of acoustic waves, unlike the present manuscript discussing scattering of electromagnetic wave. Additionally, the present manuscript discusses practical feats and limitations of sub-resonant scattering (see the distinctions made in the 2nd half of the introduction), whereas the mentioned paper focuses on scattering at resonances, using simulated scenarios. 

Reviewer 3 Report

Comments and Suggestions for Authors

In this contribution, a bi-static scattering measurements at millimeter-wave frequencies are applied to characterize sub-resonant dielectric spheres of sub-wavelength size. In particular, size and relative permittivity are extracted simultaneously from measurements at Ka-band (26.5 - 40 GHz). However, a point require further clarification.

-) In Figure 8 is shown the role of the frequency bandwidth for the extraction of the parameters. In general, the frequency band affect positively the inverse scattering procedure, such as the extraction parameters of a sphere. However, the analysis in not exhaustive. Accordingly, the authors should provide a more detailed analysis on the role of frequency, such as analyzing the bandwidth effect and the center frequency location.

Author Response

Reviewer 3
“… a point require further clarification: In Figure 8 is shown the role of the frequency bandwidth for the extraction of the parameters. In general, the frequency band affect positively the inverse scattering procedure, such as the extraction parameters of a sphere. However, the analysis in not exhaustive. Accordingly, the authors should provide a more detailed analysis on the role of frequency, such as analyzing the bandwidth effect and the center frequency location.”

Answer:
The role of the frequency bandwidth is qualitatively discussed in section 2.3 (page 7, related to Fig. 8). A more quantitative analysis of how much measurement noise could be tolerated at a given frequency bandwidth (or, considering a specific centre frequency location) seems very specific to our measurement setup/equipment and to the used algorithms and will likely be very intricate. Therefore, in the present manuscript, we restrict ourselves to the shown, rather qualitative discussion.

Reviewer 4 Report

Comments and Suggestions for Authors

Review of the manuscript “Characterization of Sub-Resonant Dielectric Spheres by Millimeter-Wave Scattering Measurements”

The manuscript presents an experimental method to retrieve the size and dielectric permittivity of spherical particles via millimeter-wave scattering measurements. The authors optimize the system parameters, including scattering angle and frequency range, to achieve their results. While the study is technically sound, several key aspects require clarification and deeper discussion to strengthen the manuscript.

Major Points:

1. The particles under study are deeply sub-wavelength, meaning their scattering is dominated by the dipole moment (α) rather than by their size (R) and permittivity (ε_r) independently. The dipole moment is given by:

α=4π R^3 (ε_r−1)/(ε_r+2)

This implies that multiple combinations of (R, ε_r) can yield the same α, leading to an inherent ambiguity in the retrieval process. The authors should:

  • Explicitly discuss this degeneracy and how their method resolves it.

  • Clarify whether multipole contributions explain the rising edges of the valley in Figure 6.

Crucially, the condition α = const explains the direction and longitudinal profile of the "valley" shape in Figure 6, as it defines a curve of degenerate (R, ε_r) solutions where the scattering response remains identical. However, the rises on the left and right edges of the valley likely stem from higher-order multipole contributions, which are not accounted for in the dipole approximation.

2. It is unclear whether the calibration factor C is frequency-dependent or constant. If constant, the manuscript should describe how it is determined across the entire frequency range.

3. The authors assume that ε_r is constant over the measured frequency range. This assumption should be justified, either with literature references or experimental evidence (e.g., prior permittivity measurements of similar materials).

Additionally, the transparency of the holder in the studied frequency range should be confirmed and discussed to rule out potential scattering artifacts.

Minor Points:

4. The waveguide orientations differ between the schematic (Figure 1) and the photograph (Figure 3). This inconsistency complicates the reader’s understanding of the experimental setup. The figures should be aligned for clarity.

5. The phrase “sub-resonant dielectric spheres of sub-wavelength size” may be redundant since sub-wavelength spheres are inherently non-resonant (or at least not in the Mie resonance sense).

6. Introduction contains references to old paper dating back to 1980s (Refs. 4-7). The manuscript would benefit from incorporation of references to modern papers describing scattering-based particle characterization, see e.g. https://doi.org/10.1039/D1AY00431J 

Author Response

Reviewer 4

“1. The particles under study are deeply sub-wavelength, meaning their scattering is dominated by the dipole moment (α) rather than by their size (R) and permittivity (ε_r) independently. The dipole moment is given by:
α=4π R^3 (ε_r−1)/(ε_r+2)
This implies that multiple combinations of (R, ε_r) can yield the same α, leading to an inherent ambiguity in the retrieval process. The authors should: Explicitly discuss this degeneracy and how their method resolves it. Clarify whether multipole contributions explain the rising edges of the valley in Figure 6.
Crucially, the condition α = const explains the direction and longitudinal profile of the "valley" shape in Figure 6, as it defines a curve of degenerate (R, ε_r) solutions where the scattering response remains identical. However, the rises on the left and right edges of the valley likely stem from higher-order multipole contributions, which are not accounted for in the dipole approximation.”

Answer:
This is a valuable addition to the paper as it further highlights the necessity to include higher-order modes in the extraction procedure. Indeed, the superposition of dipole moments (that is, more than one moment) allows for the simultaneous, extraction of separate R and ε_r. A discussion of this point is added to the revised manuscript on page 8: too long to repeat here, equations (7), (8) and two paragraphs there (page 8, lines 213-228) are added. In order to comply with the notation of the manuscript, the phenomenon is described in terms of the dipole coefficient a_1 instead of the dipole moment α.

“2. It is unclear whether the calibration factor C is frequency-dependent or constant. If constant, the manuscript should describe how it is determined across the entire frequency range.”

Answer:
The calibration factor C is frequency dependent and is calculated at each frequency point. The text below equation 3 (page 5, line 126) now reads as follows: “The frequency-dependent calibration factor C is found …”

“3. The authors assume that ε_r is constant over the measured frequency range. This assumption should be justified, either with literature references or experimental evidence (e.g., prior permittivity measurements of similar materials).”

Answer:
There is evidence in the literature supporting an only very small frequency dependence of permittivity of alumina ceramic. In the revised manuscript, a reference is added to the text: see page 6, lines 171-172: “This assumes that the permittivity is constant over the considered frequency range (in accordance with [36]).”.

“Additionally, the transparency of the holder in the studied frequency range should be confirmed and discussed to rule out potential scattering artifacts.”

Answer:
For the sphere holding structure, we use a low-permittivity plastic foam-like material (Rohacell, permittivity approx. 1.08), while in other papers (such as [15]), Polystyrene (permittivity 2.5) sockets are used. We use a socket in the shape of a pyramid socket. Thus, we believe that the influence of this socket is minimized. Any influence related to far-field interaction (scattering by the socket, specular reflections or so) are calibrated/subtracted away (considered as a part of the “direct coupling” signal path) by the measurement procedure. We tried (by simulations and measurements) but could not find any near-field related interaction of the sphere with the socket.  

“4. The waveguide orientations differ between the schematic (Figure 1) and the photograph (Figure 3). This inconsistency complicates the reader’s understanding of the experimental setup. The figures should be aligned for clarity.”

Answer:
A coordinate system is added to Fig. 3 in accordance with the one in Fig. 1. The drawing in Fig. 1 is modified to show the flanged waveguides in a 3D view. Note that both figures, Fig. 1 and Fig. 3, just show example constellations for the investigated in-plane scattering (in the xz-plane) with different polarizations (in-plane pol, normal pol) and for different directions (theta).  

“5. The phrase “sub-resonant dielectric spheres of sub-wavelength size” may be redundant since sub-wavelength spheres are inherently non-resonant (or at least not in the Mie resonance sense).”

Answer:
The wavelength in the term “sub-wavelength size” refers to the free-space wavelength, lambda_0 (for vacuum/air as background medium). One can either define “sub” as diameter smaller than wavelength (D<lambda_0), or ka<1 (that is, D<lambda_0/pi). In either case, particles of sufficiently high permittivity of sub-wavelength size can show resonances. For example, the alumina spheres used here (ε_r approx 10) are resonant at approximately x = ka = 0.95 < 1, thus at sub-wavelength size. For clarity, the manuscript uses the terminology of “sub-resonant dielectric spheres of sub-wavelength size”.

“6. Introduction contains references to old paper dating back to 1980s (Refs. 4-7). The manuscript would benefit from incorporation of references to modern papers describing scattering-based particle characterization, see e.g. https://doi.org/10.1039/D1AY00431J”

Answer:
The recommended paper is now referred to in the introduction (page 2, line 38, ref [9]).

Round 2

Reviewer 3 Report

Comments and Suggestions for Authors

It is well known that the frequency bandwidth in inverse scattering problem improves the performance for the extraction of the parameters. Obviously, the noise is always present and it limits the accuracy of the estimated parameters. However, my request was to address how frequency impacts performance achievable with your proposed method.

Author Response

“It is well known that the frequency bandwidth in inverse scattering problem improves the performance for the extraction of the parameters. Obviously, the noise is always present and it limits the accuracy of the estimated parameters. However, my request was to address how frequency impacts performance achievable with your proposed method.”

Referring to "how frequency impacts performance achievable with your proposed method", we answer as follows:

 - Our investigations are using frequencies in 26.5-40 GHz range. Since microwave scattering is a passive, linear process, everything scales with wavelength and the results apply quantitatively to other frequencies of the microwave and mm-wave range.

 - At higher frequency, however, experimental efforts are much higher, because mechanical tolerances are tighter and the transmission measurement becomes more expensive (cheap equipment will have reduced signal-to-noise ratio). Therefore, we choose Ka-band.

 - Fig. 8 indicates the effect of bandwidth. The second paragraph below Fig. 8 discusses these effects.

Sincerely,

Jan Hesselbarth

Reviewer 4 Report

Comments and Suggestions for Authors

The authors have replied to my comments.

Author Response

“The authors have replied to my comments.”

We thankfully acknowledge this comment.

Sincerely,

Jan Hesselbarth